

# Antibacterial immune functions of subadults and adults in a semelparous spider

Zoltán Rádai, Péter Kiss, Dávid Nagy and Zoltán Barta

MTA-DE Behavioural Ecology Research Group, Department of Evolutionary Zoology, University of Debrecen, Debrecen, Hungary

## ABSTRACT

Although capacity to mount an efficient immune response plays a critical role in individuals' survival, its dynamics across ontogenetic stages is still largely unexplored. Life stage-dependent variation in the encountered diversity and prevalence of parasites were proposed to contribute to stage-dependent changes in immunity, but differences in life history objectives between developmental stages may also lead to stage-specific changes in efficiency of given immune mechanisms. The reason for this is that juveniles and subadults are unable to reproduce, therefore they invest resources mainly into survival, while adults have to partition their resources between survival and reproduction. The general trade-off between somatic maintenance and reproductive effort is expected to impair immune function. Especially so in semelparous organisms that only reproduce once throughout their lifetime, hence they do not face the trade-off between current and future reproduction. We hypothesised that in a semelparous species individuals would be characterised by decreased investment into somatic maintenance after maturation, in order to maximise their reproductive output. Accordingly, we predicted that (1) elements of somatic maintenance, such as immunity, should be relatively weaker in adults in comparison to subadults, and (2) increased reproductive investment in adults should be associated with lower immune efficiency. We quantified two markers of immunity in subadult and adult individuals of the semelparous wolf spider *Pardosa agrestis* (*Westring, 1861*), namely bacterial growth inhibition power and bacterial cell wall lytic activity. We found that subadults showed significantly higher cell wall lytic activity than adults, but the two life stages did not differ in their capacity to inhibit bacterial growth. Also, we found weaker immune measures in mated females compared to virgins. Furthermore, in mated females bacterial growth inhibition power was negatively associated with fecundity.

## INTRODUCTION

The capacity to effectively respond to antigens to minimize fitness costs of infection (*Owens & Wilson, 1999*) is a critically important element of somatic maintenance. Although its efficiency is highly correlated with fitness (*Rolff, 2002*; *Viney, Riley &*

Corresponding author
Zoltán Rádai,
radai.zoltan@science.unideb.hu

*Buchanan, 2005*), considerable variation persists in the efficacy of immune functions in natural populations (*Brinkhof et al., 1999*; *Norris & Evans, 2000*; *Calsbeek, Bonneaud & Smith, 2008*). In the last two decades, immunoecology aimed at shedding light on the causes and consequences of such diversity. Until recently numerous studies were carried out in this subject (reviewed in: *Sheldon & Verhulst, 1996*; *Lochmiller & Deerenberg, 2000*; *Harshman & Zera, 2007*; *Cotter et al., 2008*; *Sadd & Schmid-Hempel, 2009*; *Crino et al., 2013*; *Palacios, Cunnick & Bronikowski, 2013*; *Park & Stanley, 2015*), mainly focusing on the variability of adult immunocompetence, and the effects of juvenile-experiences (such as immune-challenge or nutritional deficiency) on adult immunocompetence (*Norris & Evans, 2000*; *Ricklefs & Wikelski, 2002*; *Jacot et al., 2005*; *Stoehr & Kokko, 2006*; *Martin, Weil & Nelson, 2007*; *DeBlock & Stoks, 2008*; *Kriengwatana et al., 2013*; *Gilbert, Karp & Uetz, 2016*). However, still surprisingly little is known about how the baseline efficiency of the immune system changes through the different life stages.

Differences in the type or density of pathogens individuals might encounter throughout their life cycle were proposed to contribute to across-stage differences in immune function (*Wilson-Rich, Dres & Starks, 2008*; *Shi & Sun, 2010*). Also, expected reproductive value of different life history actions may change with the progression of ontogeny (e.g. investment in reproduction might be more valuable in mature than in immature individuals), so optimal investment into somatic maintenance, and hence into immunity, might also show stage-specific changes (*Shi & Sun, 2010*; *Giglio & Giulianini, 2013*). Indeed, juveniles and subadults can't reproduce, therefore they would be expected to invest resources mainly into somatic maintenance, while adults have to partition their resources between maintenance (and hence survival) and reproduction.

Stage-specific characteristics of organisms' life histories can strongly depend on their reproductive strategies. In semelparous organisms, individuals have only one chance to reproduce during their lifetime, and hence cannot trade off between current and future reproduction. Consequently, after maturation they would be expected to prioritise reproduction over somatic maintenance during the production of costly gametes, which starts shortly after maturation. We expect to see this prioritisation to be pronounced in semelparous species, because the production of gametes in sufficient amount (and quality) shortly after maturation has critical impact on lifetime reproductive success, as there will be no future reproductions during which an individual might increase its lifetime reproductive success. In such a scenario, we would expect to see that individuals of a semelparous species invest less into somatic maintenance (and hence into costly elements of immunity as well) in adulthood relative to the somatic maintenance during preadult development.

In this study, we evaluated two proxies of immune capacity in subadult and adult individuals of the wolf spider *Pardosa agrestis* (*Westring, 1861*), and quantified reproductive investment of females using the number of their offspring. Individuals of *P. agrestis* mature and reproduce during spring and early summer, and offspring either overwinter and reproduce in the next year, or during late summer and autumn, i.e. there are two separate adult peaks throughout the year (*Kiss & Samu, 2005*; *Rádai, Kiss & Samu, 2017*). In general, gamete production starts shortly after maturation in spider females

(*Foelix, 2011*). Although little is known about sperm production in males, generally males also start gamete production shortly after the final moulting, and sperm is then excreted and stored in the pedipalps until mating. Adults reproduce shortly after maturation, not long after which males perish. Females usually lay one to three cocoons after mating, and carry the hatched spiderlings for a short period of time, then also perish. We chose *P. agrestis* as our model organism because it is a quite abundant semelparous spider in Europe, and is relatively easy to collect and study in laboratory (*Rádai, Kiss & Barta, 2017*; *Rádai, Németh & Barta, 2018*). This semelparous spider species, by definition, reproduces once during its lifetime (*Kiss, 2003*). Therefore, based on the argument outlined above, one would expect individuals after maturation to invest less in somatic maintenance compared to premature individuals.

For proxies of immunocompetence we measured the efficiency of bacterial growth inhibition and cell wall lytic activity of tissue homogenates prepared from the spiders' abdomens. In the bacterial growth inhibition assay the availability of antimicrobial peptides is assessed, which are released in the presence of bacteria or fungi at the site of infection, as part of the spiders' humoral immune response (*Gao et al., 2005*). These molecules mainly compromise the integrity of cell wall and cell membrane of the invading microbes (*Brogden, 2005*). Antimicrobial peptides are usually stored in granulocytes (specialised hemocytes), which are produced in the heart wall, from prohemocytes, but can be found in fat bodies as well (*Kuhn-Nentwig & Nentwig, 2013*). Lytic activity of lysozymes, is also important part of the humoral immune response. Being glycoside hydrolases, lysozymes conduct the lysis of the peptidoglycans in the bacterial cell wall (*Sharon, 1967*). Although it is unknown where lysozymes are produced and stored in spiders, the hemolymph contains them in considerable quantities (*Ahtiainen et al., 2005*). Therefore the homogenate sample of the spider abdomen is likely adequate to assess lysozyme activity, as the largest hemolymph vessel (i.e. heart) in the spiders' body is located in the abdomen. Because these biomolecules are not only produced during infection but are also present in the hemolymph and some cells of healthy individuals, measuring the growth inhibition and lytic activity of samples (which are proportional to the concentrations of corresponding molecules in the sample, see *Castella, Christe & Chapuisat, 2010*) informs us about the amount of the deployable molecules, and hence the level of investment into the maintenance of these immune functions (*McKean & Lazzaro, 2011*).

Although bacterial growth inhibition assay and lytic activity assay assess similar immunological functions (e.g. disruption of the cell wall integrity and production of bacteria), bacterial growth inhibition is a result of the effect of several types of peptides, with various mechanisms acting on a wide range of microbes, i.e. are expected to be effective against microbes in general when released simultaneously. Indeed, a number of bactericide and fungicide antimicrobial peptides were identified in arachnids, which collectively are thought to be efficient in the elimination of a wide range of microbes (see *Bednaski et al., 2015*). Notably, lytic activity strictly assesses the activity and availability of lysozymes, i.e. a type of enzymes with the sole function of disrupting the cell wall of Gram positive bacteria. Therefore, the former assay represents a more general assay on the potential capacity to prevent (or halt) microbial invasion, whereas the latter

informs about the potential capacity to neutralise Gram positive bacteria susceptible to lytic enzymes.

Using the above described assays we investigated whether subadult and adult individuals of *P. agrestis* differ in the bacterial growth inhibition power and cell wall lytic activity, i.e. in the maintained capacity to mount either a complex, or a simpler, immediate immune response against microbial infection. For these tests we used males and females as well, in order to see whether the costs of maturation interfere with adult immunity in both sexes. Sexes might differ in how they trade off key life history traits due to their differences in the ways they can maximise reproductive output (*Lawniczak et al., 2007*; *Vincent & Gwynne, 2014*). Additionally, in females collected from natural habitats we assessed how reproductive status (whether a female was virgin or not) and fecundity (number of offspring) were associated with the immune parameters we used. We predicted to see stronger immune parameters in subadult specimens, and expected mating and egg laying to negatively affect the tested females' immune measures. Similarly, due to the energetic trade-offs between somatic maintenance and reproduction, we expected to see negative associations between fecundity and the immune parameters.

## MATERIALS AND METHODS

### Spiders

We collected spiders on three occasions. Firstly, we have established a laboratory-reared population, in which specimens originated from nine cocoon carrying females. These females were collected in May of 2016, from two closely located habitats: an uncultivated plot next to a maize field (between Hajdúszoboszló and Nádudvar, Hungary; 47°26′56.3″N 21°18′00.2″E), and the edge of a maize field (between Hajdúszoboszló and Nagyhegyes, Hungary; 47°28′31.2″N 21°21′57.3″E). These habitats were quite close to one another (ca. five km linear distance), established on an alluvial plain, i.e. with minimal differences were expected in habitat characteristics. After spiderlings hatched in May and June, we separated 58 specimens, and placed each of them separately in plastic cups with a floor area of approx. 25.5 cm$^2$. Spiderlings were provided water (wet cotton wool) and food (springtails for the first few moults, then fruit flies for larger spiders) ad libitum. Cups were checked for exuvia (i.e. moulting) every 2 days, and the dates of hatching and moulting were recorded for each spider. Spiders were reared until December, 2016. Prior to the preparation of samples, each spider was categorised to be either as subadult (undeveloped, but recognisable genitals; observed in penultimate specimens one moult prior to maturation), or as adult (fully developed genitals). Henceforth, we refer to spiders that were one moult prior to maturation (i.e. penultimate) as 'subadult'. We chose to work with subadults rather than juvenile instar larvae because the small sizes of juveniles makes it rather difficult to use the established methods reliably. Following the rearing period, we were able to work with 38 spiders, of which 27 were subadults (11 females and 16 males) and 11 were adults (seven females and four males). All laboratory reared spiders used in the immune assays were virgins.

Secondly, we have collected subadult females and adult cocoon carrying females at spring and late summer in 2017 from the habitats described above. Subadult females were

reared until maturity in laboratory. This way we had virgin and mated females as well for our tests. In total we were able to work with 154 females, among which there were 37 virgin and 66 mated females from spring, and 23 virgin and 28 mated females from late summer. Because some of the cocoons from the mated females either enveloped dead embryos, were parasitized by *Idris flavicornis* (Hymenoptera: Scelionidae) wasps (see *Rádai et al., 2018*), or were used for other tests, we had fecundity data from 25 spring and 28 late summer females.

## Sample preparation

Laboratory-reared spiders were used for sample preparation in December, 2016. Females collected from natural habitats were used for sample preparation in May and August, 2017 in the case of spring and late summer females, respectively.

Bacterial growth inhibition capacity and cell wall lytic activity assays were carried out according to *Castella, Christe & Chapuisat (2010)*. For both methods the samples were prepared as follows. Spiders were killed at −20 °C. The abdomen of the spiders was separated from the prosoma by cutting the pedicel. We decided to use full abdomens because the heart and fat bodies, which are known to have central role in the production of immune cells, can be found in the abdomen. Also, a large portion of hemolymph resides in the abdomen, that contains immune cells and lysozyme. Furthermore, the lung-openings and intestines are also found in the abdomen, which comprise a considerable contact surface for potential microbial invasion, so it would be expected that immune parameters of neighbouring tissues faithfully represent the spiders' readiness to keep microbes at bay.

Abdomens were homogenised on ice, in 50 μl PBS (ca. 4 °C), and stored on ice until centrifugation. Centrifugation was carried out at 10,000×$g$, at 4 °C for 10 min. After centrifugation, 20 μl from the supernatant per sample were collected separately for bacterial growth inhibition and cell wall lytic activity tests, then stored at −20 °C until measurements. Prosomas were stored in 96% ethanol on −20 °C for later size measurements, which was done with the prosomas photographed on millimetre graph paper, then measuring with ImageJ (version 1.46a, *Schneider, Rasband & Eliceiri, 2012*).

Samples from 2016 were tested in January, 2017, and samples from 2017 were tested in December, 2018. During the bacterial growth inhibition tests we always used freshly cultured bacteria (see below) from the same laboratory strain. Also, in the cell wall lysis tests we used the same type of commercially available cell wall residues (see below) in both test sessions.

## Bacterial growth inhibition assay

We applied two μl from each tissue homogenate sample on the surface of a thin layer of agar-based medium (using six ml of agar in each petri dish), in duplicates from each spider. Duplicates were placed in the same petri dish in the case of all samples, and in one petri dish 10 samples (i.e. five duplicates) were applied. The medium contained one g of bacto-tryptone, one g NaCl, 0.5 g yeast extract and one g agar in 100 ml distilled water. Also, prior to pouring the agar into petri dishes, 50 μl from a $2.6 \times 10^8$ cells ml$^{-1}$ suspension of *Micrococcus luteus* (obtained from Tamás Emri's lab culture at the Department of

Microbial Biotechnology, University of Debrecen) was added and mixed carefully. We used *Micrococcus luteus* because of its availability, and because it is frequently present in soil (*Biskupiak et al., 1988*; *Sims, Sommers & Konopka, 1986*); as *P. agrestis* is an epigeic species, it is expected to frequently encounter *Micrococcus luteus* straints. Petri dishes with the samples applied were incubated at 30 °C for 24 h. Samples containing antimicrobial peptides inhibit bacterial growth, therefore empty zones (so called inhibition zones) will persist on the bacterial culture where antimicrobial peptide containing samples were applied. The area of inhibition zones is proportional to the antimicrobial peptide content of the samples. Following the incubation, all petri dishes were photographed, and the areas of inhibition zones were measured using ImageJ (see below).

## Bacterial cell wall lytic activity assay

We prepared a simple agar-based medium (one g agar in 100 ml distilled water) containing five mg ml$^{-1}$ *Micrococcus lysodeikticus* (lyophilized *Micrococcus lysodeikticus*, ATCC No. 4698, Sigma-Aldrich; note that *Micrococcus lysodeikticus* and *Micrococcus luteus* are considered to be the same group of bacteria, e.g. *Litman, 1968*) cell wall residues homogeneously distributed, and 0.6 mg ml$^{-1}$ ampicillin sodium. This strain was used because of its commercial availability. The basic principle of this method is that samples containing lysozyme will produce transparent clearing zones in the opaque layer of cell wall containing agar, and the area of clearing zones is proportional to the lysozyme concentration of the sample. We poured five ml of this medium in each petri dish and applied two μl from each sample on the surface of the cell wall agar, in duplicates from each spider. Duplicates were placed in the same petri dish in the case of each sample, and in one petri dish 10 samples (i.e. five duplicates) were applied. Petri dishes were then incubated at 30 °C for 24 h. Following the incubation, all petri dishes were photographed, and the areas of clearing zones were measured using ImageJ (see below).

## Statistical analyses

Statistical analyses were performed using the R statistical software (version 3.0.2, *R Development Core Team, 2014*).

Because the cut-off point of rearing period was the same for all spiders, we checked whether there was a significant difference in the time interval from hatching to sample preparation between subadults and adults, by using Welch's *t*-test on developmental times (i.e. time from hatching to end of rearing period). A lack of significant difference would imply that spiders had approximately the same time to develop, therefore those specimens that reached maturity had a more rapid somatic development than those that couldn't reach maturity (i.e. remained subadults). Conversely, if adult spiders were characterised by significantly longer developmental time it would mean that they hatched more early and had more time to mature, hence their maturation success during the rearing period is not a direct result of faster development. Additionally, we used Chi-squared ($\chi^2$) test to see if there are significant differences in the number of adults and subadults between mother spiders.

## Quantifying bacterial growth inhibition and cell wall lytic power

Bacterial growth inhibition and cell wall lytic capacities were quantified as follows. Measured areas of inhibition- and clearing zones (for bacterial growth inhibition and cell wall lytic activity, respectively) were given in pixel count. Photographing the petri dishes was done at different times, hence on the distance between the camera and the petri dishes varied slightly between images. In order to render inhibition/clearing zone values comparable between petri dishes, we divided their area by the area of the given petri dish (all petri dishes were the same size). This resulted in values representing the relative size of inhibition and clearing zones, proportional to the total area of the given petri dish (hereafter relative zone areas). Relative zone areas were divided by the cube of prosoma width to calculate inhibition- and clearing zones corrected for body volume. As whole abdomens of different sized spiders were homogenized in the same volumes of buffer, the relative zones corrected with the cube of prosoma width estimated the inhibition- and lytic power of samples per mm$^3$ of body volume. In order to aid convergence of model fitting, and interpretation of results after analyses, the variables representing bacterial growth inhibition power and cell wall lytic power (i.e. size-corrected relative zone areas) were centred at zero (by subtracting the mean of the given variable from each value) and rescaled (by dividing all values by the standard deviation of the given variable). Because the response variables were divided by their standard deviation, regression coefficients coincide with the effect sizes (Cohen's d).

## Developmental stage differences in immune markers

To quantify growth rate we used a variable calculated as dividing the cube of prosoma width by developmental time (i.e. number of days from hatching to the end of rearing). We analysed our data from an animal-model approach (*Wilson et al., 2010*) using the R-package 'MCMCglmm' (*Hadfield, 2010*), applying a multi-response mixed effects model. We chose this modelling approach because by fitting only one model we are able (a) to estimate regression coefficients (and the coefficients' credibility) for predictor variables on multiple response variables simultaneously, (b) to control for random effects (such as pseudo-replication from repeated measurements) and hence to estimate measurement repeatability, and (c) to estimate coefficients (and their credibility) representing association between the responses, while controlling for a number of fixed- and random-effects. In this specific model we used data only from laboratory-reared spiders.

The multi-response model was built as follows: bacterial growth inhibition power and cell wall lytic activity were response variables, and, for both response variables, life stage, sex and growth rate were predictors. We did not include interactions between predictors in order to avoid over-parameterization. In the model we used the ID number of spiders as random factor to control for repeated measurements, ID of petri dishes to control for between petri dish error, and the ID number of the spiders' mother to control for kinship between spiders. Variance structures of the model were defined to estimate random intercept- and residual covariance between responses. The former reflects the covariation between random intercepts estimated for the responses, practically
representing correlation between responses. A positive random intercept covariance indicates that those observations with higher-than-average values in the first response variable tend to have similarly higher-than-average values in the second response. Residual covariance represents the covariation between residuals of the responses. For example, a negative residual covariance might indicate the presence of a trade-off between the two responses, in the sense that those individuals investing too much into one trait will be characterised by lower values in the second response. Note that positive random intercept covariance and negative residual covariance can occur at the same time, hinting at that good quality specimens are able to invest more-than-average into both responses, but extremely high investment into the first will not enable similarly high investment to the second (e.g. due to the limited nature of used resources; see *Hadfield, 2018*).

The model was run for 115,000 iterations (out of which the first 15,000 was discarded as 'burn-in'), with a thinning interval of 20, resulting in a nominal sample size of 5,000 in the posterior distributions. Following model fitting we have estimated the precision of the used methods as the repeatability of the within-individual measurements, by calculating the proportion of between individual variance to the sum of between individual and residual variance. Low values of this measure indicate low precision, while high values indicate reliable measurements of high precision.

In the results we report parameter estimates (coefficients) as the modal (most frequent, i.e. most probable) values from the posterior distributions. Also, Bayesian credible intervals are presented as the lower ($HPD_{lower}$) and upper ($HPD_{upper}$) cut points of the 95 % highest posterior density intervals.

## Association between reproductive status, fecundity and immune markers

We have fitted a multi-response model, using data from the females collected from the natural habitats in 2017. In this model the response variables were bacterial growth inhibition power and cell wall lytic activity, and predictors were reproductive status (virgin vs. mated), season of collection (spring vs. late summer), and the interaction between status and season. We defined ID number of spiders, and ID number of petri dishes as random effects to control for pseudo-replication and between-petri error. Similarly to the above described model, we fitted the model in a manner so that random intercept- and residual covariances were estimated. The model was run for 115,000 iterations (out of which the first 15,000 was discarded as 'burn-in'), with a thinning interval of 20, resulting in a nominal sample size of 5,000 in the posterior distributions. Repeatability of immune measurements was calculated the same way as for measurements on laboratory-reared spider samples.

To test how efficiency of bacterial growth inhibition and cell wall lytic activity were associated with fecundity in mated females, we fitted a classical (frequentist) linear regression model. In this model fecundity was the response variable, and the assessed immune measures were predictor variables. We also included season of spider collection as a confounding variable, and its interaction with both immune markers. Because a substantial number of samples produced no measurable inhibition and/or clearing zones,

we additionally fitted binomial generalised regression models, in order to test whether fecundity was associated with the probability of producing measurable (i.e. higher than zero) zones. In these models we specified binary variables as responses, indicating whether inhibition zones in bacterial growth inhibition assay, and clearing zones in cell wall lytic assay, were measurable or not. Predictor variables were fecundity, season of collection, and the interaction term between the two. Since in previous models season appeared to affect the response variables in the above described frequentist models, we estimated variance inflation factor (VIF) for the models, using the R-package 'fmsb' (*Nakazawa, 2018*); since in the frequentist models VIF values were low (below 2.5), we assessed that the correlating independent variables do not cause substantial multicollinearity. Additionally, because for each female we had two measurements of the immune markers in the data used by the afore described two frequentist models, but only had one fecundity value, we used the averaged bacterial growth inhibition and cell wall lytic values for each female.

In the results of Bayesian models we report $P_{MCMC}$-values as the proportion of MCMC samples not crossing zero in the posterior distribution. These values practically represent the probability that (based on the data) the posterior mean of the regression coefficient is zero. Using the $P_{MCMC}$ values we specified the threshold for significance to be 0.05. Note that assessment of significance using this threshold coincides with the usage of 95% credible intervals, i.e. statistical significance can be assessed based on whether the credible intervals include zero. That being said, we decided to still report $P_{MCMC}$ values to ease interpretation of results for those not familiar/comfortable with Bayesian statistics.

## RESULTS

### Developmental stage differences in immune markers

Spiders that reached maturity by the end of the rearing period on average hatched earlier than those that only reached subadult stage ($t_{31.6} = -4.38$, $P < 0.001$), indicating that individuals became adults because they had more time to develop rather than having much more rapid development. Also, the Chi-squared test showed no significant difference in the number of adults and subadults between mothers ($\chi^2_8 = 4.90$, $P = 0.768$).

There was no significant difference in bacterial growth inhibition power between subadult and adult spiders (Figs. 1A and 1B), and growth rate did not affect bacterial growth inhibition (Table 1). Males showed marginally significantly lower values of bacterial growth inhibition than females (Fig. 1B). Adults showed significantly lower cell wall lytic activity than subadults (Figs. 1C and 1D), but there was no difference between sexes, and growth rate did not have significant effect on cell wall lytic activity (Table 1). The estimated precision of the bacterial growth inhibition measurements was quite high ($R = 0.87$, $HPD_{lower} = 0.71$, $HPD_{upper} = 0.93$). Precision of cell wall lytic activity measurements was moderately good ($R = 0.63$, $HPD_{lower} = 0.39$, $HPD_{upper} = 0.80$).

We found no significant random intercept- (random intercept covariance = $-0.06$, $HPD_{lower} = -0.22$, $HPD_{upper} = 0.13$, $P_{MCMC} = 0.672$) or residual covariance (residual covariance = $0.004$, $HPD_{lower} = -0.03$, $HPD_{upper} = 0.05$, $P_{MCMC} = 0.560$) between bacterial growth inhibition and cell wall lytic activity.
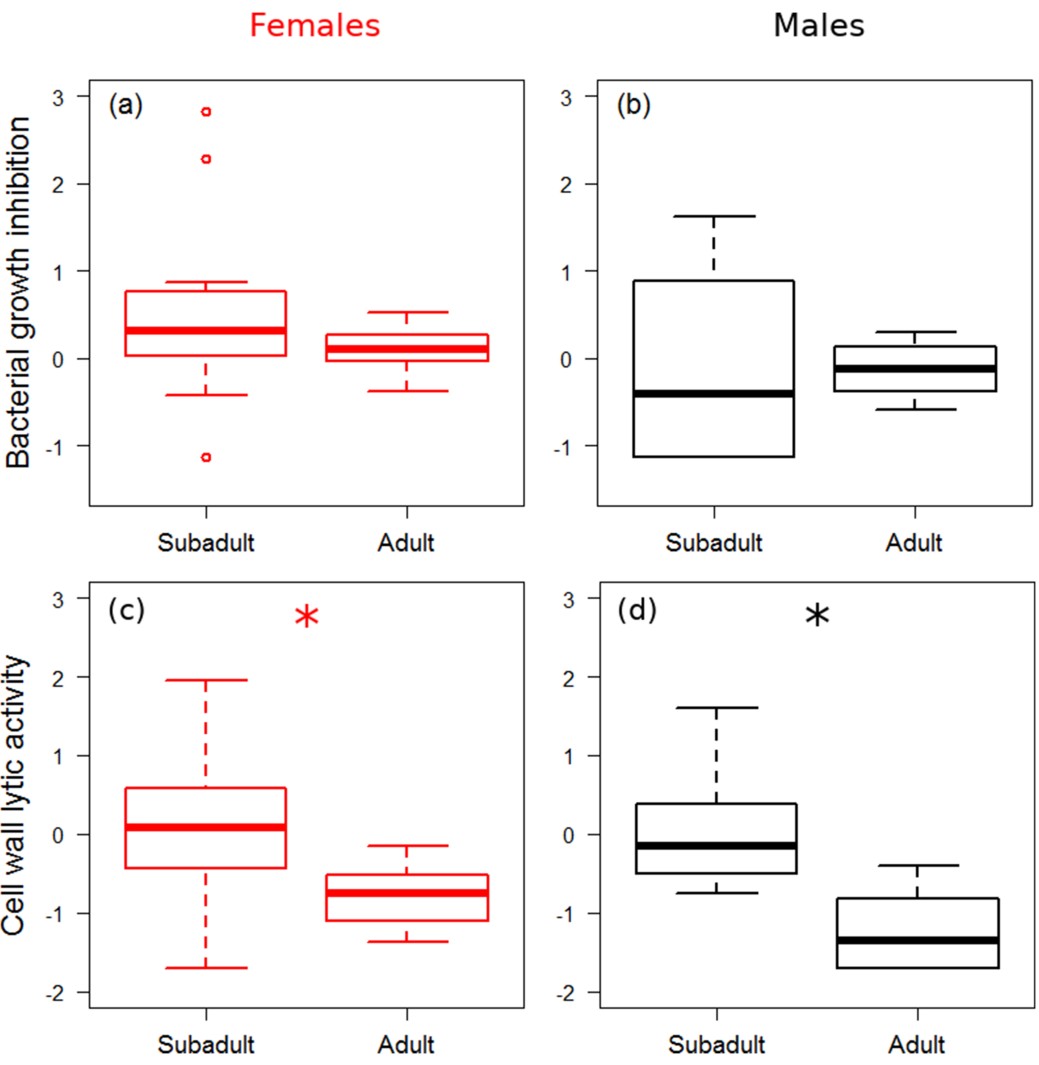

**Figure 1 Immune function across ontogenetic stages.** Box plot visualisation of bacterial growth inhibition (A and B) and cell wall lytic activity (C and D) of females (A and C) and males (B and D) of laboratory-reared spiders ($N = 38$). Whiskers visualise minimum and maximum ranges, within which boxes represent interquartile range between first and third quartiles; horizontal solid lines show median of the given value distribution; individual circles show outlier values. Asterisks mark significant effect or difference ($P \leq 0.05$).

## Association between reproductive status, fecundity and immune markers

The interaction between season and reproductive status in bacterial growth inhibition power was not significant (Coefficient: 0.429, $HPD_{lower} = -0.298$, $HPD_{upper} = 1.121$, $P_{MCMC} = 0.245$), and mated females showed significantly lower levels of bacterial growth inhibition (Figs. 2A and 2B; Table 2). Females that were collected during spring showed significantly lower levels of bacterial growth inhibition than females from late summer (Fig. 2) (Coefficient: $-1.220$, $HPD_{lower} = -1.784$, $HPD_{upper} = -0.681$, $P_{MCMC} < 0.001$).

The effect of interaction between reproductive status and season was significant on the cell wall lytic activity (Coefficient: 1.064, $HPD_{lower} = 0.412$, $HPD_{upper} = 1.757$,

**Table 1 Fixed parameter estimates from the multi-response model on the effect of developmental stage and sex on immune markers.** For categorical predictors (ontogenetic stage and sex) the contrast parameter is presented (i.e., average difference between two contrasted groups; subadult (S) to adult (A) in Stage, female (F) to male (M) in Sex). Asterisks mark significant effect or difference ($P \leq 0.05$).

| Response | Predictors | Coefficient | HPD$_{lower}$ | HPD$_{upper}$ | $P_{MCMC}$ |
|---|---|---|---|---|---|
| Bacterial growth inhibition | Stage (S–A) | 0.27 | −0.30 | 0.86 | 0.304 |
| | Sex (F–M) | −0.50 | −1.03 | 0.03 | 0.067 |
| | Growth rate | 0.71 | −5.23 | 6.27 | 0.790 |
| Cell wall lytic activity | Stage (S–A) | 1.04 | 0.57 | 1.60 | <0.001* |
| | Sex (F–M) | −0.17 | −0.66 | 0.33 | 0.489 |
| | Growth rate | 1.46 | −4.41 | 7.12 | 0.620 |

$P_{MCMC}$ = 0.004): while there was no significant difference in cell wall lytic activity between virgin and mated females collected during spring (Coefficient: −0.216, HPD$_{lower}$ = −0.621, HPD$_{upper}$ = 0.219, $P_{MCMC}$ = 0.362), among the females collected in late summer mated specimens showed significantly lower cell wall lytic activity than virgin females (Coefficient: −0.859, HPD$_{lower}$ = −1.237, HPD$_{upper}$ = −0.486, $P_{MCMC}$ < 0.001; Figs. 2C and 2D).

The estimated precision of both the bacterial growth inhibition ($R = 0.84$, HPD$_{lower}$ = 0.78, HPD$_{upper}$ = 0.88) and cell wall lytic activity measurements were high ($R = 0.84$, HPD$_{lower}$ = 0.77, HPD$_{upper}$ = 0.88). There was no significant random intercept (random intercept covariance = 0.009, HPD$_{lower}$ = −0.078, HPD$_{upper}$ = 0.087, $P_{MCMC}$ = 0.918) or residual covariance (residual covariance = −0.007, HPD$_{lower}$ = −0.021, HPD$_{upper}$ = 0.008, $P_{MCMC}$ = 0.297) between bacterial growth inhibition and cell wall lytic activity, indicating no measurable association between them.

In the linear regression model, neither of the interaction terms were significant (between bacterial growth inhibition and season: Coefficient = −2.564, SE = 3.213, $t = −0.80$, $P = 0.429$; between cell wall lytic activity and season: Coefficient = −0.699, SE = 2.773, $t = −0.25$, $P = 0.802$), therefore we excluded them from the final model. Bacterial growth inhibition power showed significant negative association with fecundity in mated females (Fig. 3A), but cell wall lytic activity did not (Fig. 3B; Table 3). Also, females collected during spring were significantly more fecund than late summer females (Table 3).

In the binomial regression model on bacterial growth inhibition the interaction between fecundity and season was not significant (Coefficient = 0.102, SE = 0.085, $t = 1.20$, $P = 0.229$), and season did not significantly affect the probability of producing a measurable inhibition zone (Coefficient = −2.503, SE = 2.834, $t = −0.88$, $P = 0.377$). Fecundity was negatively associated with this probability, meaning that samples from more fecund females were more likely to produce no bacterial growth inhibition at all (Coefficient = −0.148, SE = 0.067, $t = −2.20$, $P = 0.028$). In the binomial regression model on cell wall lytic activity the interaction between fecundity and seasonality was significant (Coefficient = 0.281, SE = 0.128, $t = 2.20$, $P = 0.028$), showing that fecundity negatively affected the probability of producing measurable clearing zones only among late summer females (Coefficient = −0.238, SE = 0.118, $t = −2.02$, $P = 0.044$). Probability of producing

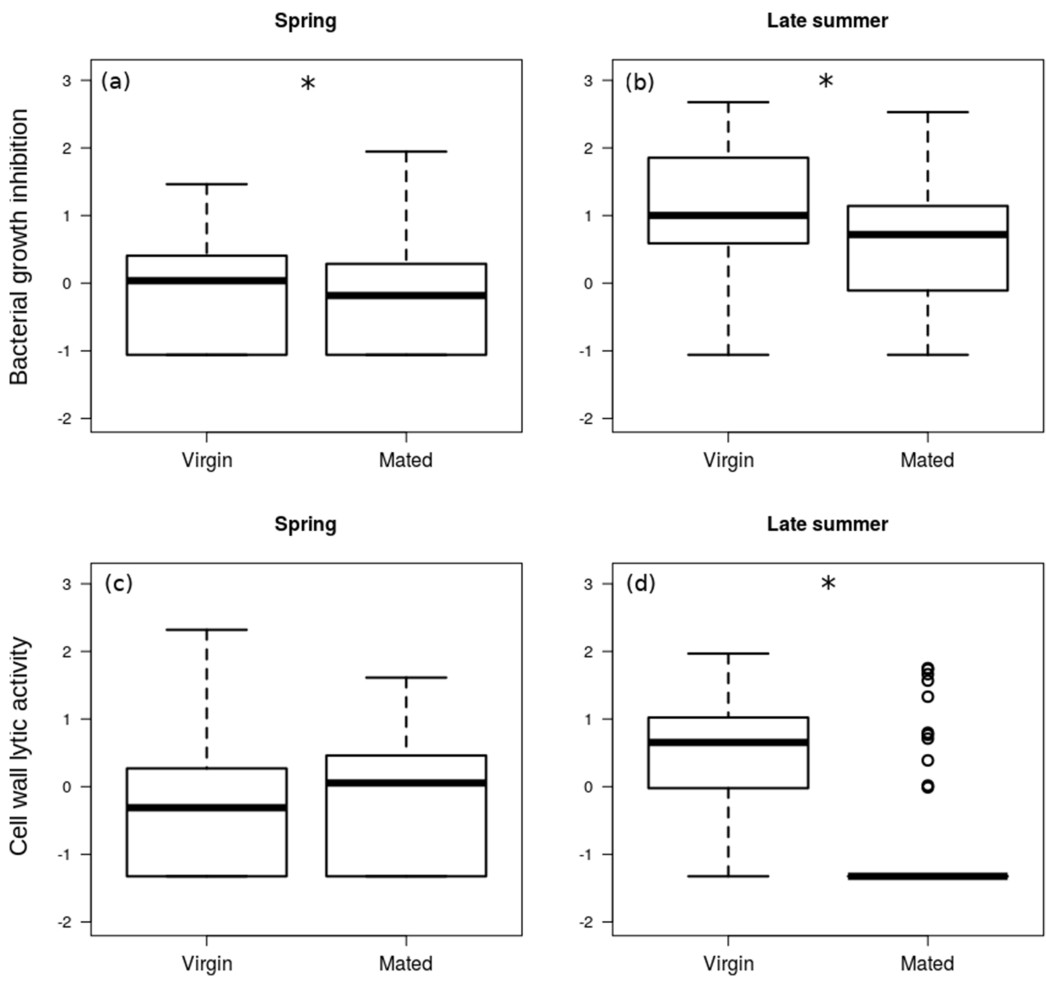

**Figure 2 Immune function in females.** Box plot visualisation of bacterial growth inhibition power (A and B) and cell wall lytic activity (C and D) of virgin vs. mated females, collected during spring (A and C; $n = 108$) and late summer (B and D; $n = 51$). Whiskers visualise minimum and maximum ranges, within which boxes represent interquartile range between first and third quartiles; horizontal solid lines show median of the given value distribution; individual circles show outlier values. Asterisks mark significant effect or difference ($P \leq 0.05$).

**Table 2 Contrasts acquired from the multi-response model on the effect of reproductive status and season on females' immune markers.** In order to make results easier to comprehend and interpret, contrasts for the given group-interactions are presented, instead of individual parameter estimates directly from the fitted model. Asterisks mark significant difference ($P \leq 0.05$).

| Response | Contrast | Coefficient | $\text{HPD}_{\text{lower}}$ | $\text{HPD}_{\text{upper}}$ | $P_{\text{MCMC}}$ |
|---|---|---|---|---|---|
| Bacterial growth inhibition | *Spring*: virgin–mated | 0.51 | 0.11 | 1.22 | 0.030* |
| | *Late summer*: virgin–mated | 0.60 | 0.13 | 1.24 | 0.019* |
| Cell wall lytic activity | *Spring*: virgin–mated | −0.16 | −0.52 | 0.45 | 0.759 |
| | *Late summer*: virgin–mated | 1.24 | 0.64 | 1.88 | <0.001* |

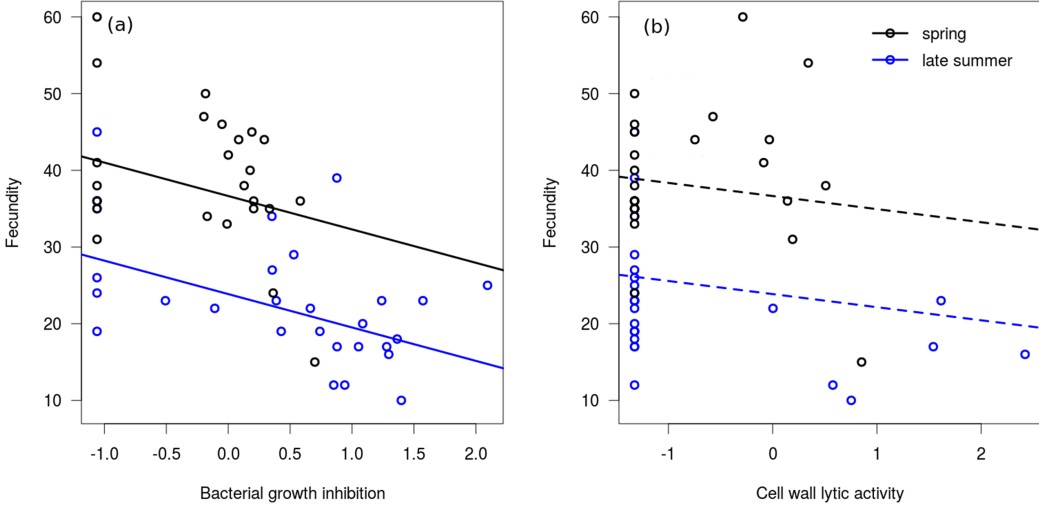

**Figure 3 Immune function and fecundity.** Association of bacterial growth inhibition (A) and cell wall lytic activity (B) with fecundity in mated females collected in spring (black circles; $n = 25$) and autumn (blue circles; $n = 28$). Lines show regression lines from the fitted linear model: solid and dashed lines represent significant ($P \leq 0.05$) and non-significant ($P > 0.05$) associations, respectively.

**Table 3 Fixed parameter estimates from the linear regression model on the association between immune markers and season with fecundity in mated females.** For the predictor variable Season the contrast coefficient is given, representing the difference of spring specimens from late summer specimens. Asterisks mark significant effect ($P \leq 0.05$).

| Predictors | Coefficient | SE | *t*-value | *P* |
|---|---|---|---|---|
| Bacterial growth inhibition | −4.15 | 1.31 | −3.16 | 0.003* |
| Cell wall lytic activity | −1.67 | 1.15 | −1.45 | 0.153 |
| Season (spring) | 12.79 | 2.36 | 5.41 | <0.001* |

measurable clearing zone was only marginally significantly lower in spring females (Coefficient = −5.595, SE = 2.986, $t = -1.87$, $P = 0.061$).

## DISCUSSION

Subadult *P. agrestis* specimens had stronger lytic activity against bacterial cell wall compared to adult individuals. This result indicates that tissue homogenate samples from subadult spiders contained higher amount of lysozyme per unit body volume, than did samples from adults. This is in accordance with our prediction based on life history theory, namely that in a semelparous species one would expect to see relatively lower efficiency of somatic maintenance in adults compared to subadults. We propose that this difference is likely due to a shift in resource allocation patterns, altering the prioritisation between somatic maintenance and reproduction. Subadults do not have fully developed, functional reproductive organs yet (*Foelix, 2011*), so their reproductive tissues are likely to have lower demand of resource income in comparison to adults. Mature individuals, however, have to invest resources into gonad development and gamete production. Consequently, a higher proportion of the acquired energy and nutrients can be assigned to somatic tissues

and to elements of somatic maintenance in subadults, while these limited resources will be partitioned into reproductive tissues and gamete production as well in adults. Producing gametes has considerable costs (*Hayward & Gillooly, 2011*; *Edward & Chapman, 2011*; *Michalik & Rittschof, 2011*), so by paying the costs of reproduction, adults are presumably able to assign fewer resources to somatic maintenance.

The decreased lytic activity in adults in comparison to subadults may be associated with lower resource allocation into this particular element of immunity, which would be in accordance with our predictions. In *Meligethes aeneus* it was shown that out of 10 screened lysozymes 4 were expressed in higher levels in juveniles in comparison to adults, although in the case of 3 lysozymes, expression levels were higher in adults (*Vogel et al., 2014*). In another study on white shrimps (*Litopenaeus vannamei*) it was found that expression level of the screened lysozyme was elevated in most of (albeit not in the last few) post-larval stages compared to adults (*Gollas-Galvan et al., 2017*), indicating a dynamic change in lysozyme availability, with a peak during the subadult stages. Notably, energetic costs of reproduction were found to be negatively linked to lytic activity in the carabid beetle *Carabus lefebvrei*, where mated individuals showed dramatically decreased lytic activity in comparison to virgin individuals (*Giglio et al., 2016*), showing that not only maturation, but reproduction too can deteriorate the efficiency of this immune function. Indeed, we found that in mated females collected during late summer, bacterial growth inhibition power and cell wall lytic activity were lower than in virgin females, indicating that the cost of reproduction is relatively higher (or, at least, more pronounced) in late summer females. Females from spring and late summer belong to two separate generations of different life cycles (*Kiss & Samu, 2005*; *Rádai, Kiss & Samu, 2017*), and it is not clear how external seasonal conditions and the spiders' intrinsic physiological characteristics contribute to the observed patterns. Albeit we acknowledge the potential importance of this aspect of the life cycle of *P. agrestis* spiders, reconciling the mechanism behind this seasonal difference in lytic activity is beyond the scope of this study.

In past studies, immune measures in subadults were found to show comparable, or even higher levels than in adults in a handful of invertebrate species. For example, although not assessed explicitly, in the study of *Vogelweith et al. (2017)* earwig (*Forficula auricularia*) juveniles seemed to have hemocyte concentrations in their hemolymph similar to that of adults, and 4th instar (penultimate) larvae even had higher hemocyte counts than adults. Also, 4th instar larvae appeared to have slightly higher total phenoloxidase activity compared to adult males, but not to females. *Piñera et al. (2013)* found that 8th and 9th instar larvae of the cricket *Acheta domesticus* had hemocyte loads and encapsulation efficiency similar to, or in some cases higher than, adults. However, juveniles had considerably lower phenoloxidase activity compared to adults. In the cricket *Gryllus texenis* males were shown to have lower phenoloxidase activity than nymphs, and their phenoloxidase activity considerably declined with age after maturation (*Adamo, Jensen & Younger, 2001*).

Surprisingly, subadults did not seem to differ from adults in bacterial growth inhibition power against *Micrococcus luteus*. This result suggests no change in the overall availability of antimicrobial peptides due to maturation. Considering studies similar to ours, however,

it seems unlikely that antimicrobial peptide levels do not change at all throughout the life cycle. For example, expression levels of antimicrobial peptides were shown to change over ontogeny in the pollen beetle *Meligethes aeneus* (*Vogel et al., 2014*). Notably, only some of the screened antimicrobial peptide transcripts were observed to decrease after maturation, indicating that not only the overall availability of these biomolecules might show ontogenetic stage-dependent shifts, but also the composition of them. How changes in the composition of available antimicrobial peptides affect the animals' capacity to overcome pathogen invasion, is a question to be considered in future studies. Notably, the absence of significant difference in bacterial growth inhibition between subadults and adults could be a result of depressed antimicrobial protein concentration in both tested life stages, in comparison to juveniles. Unfortunately, in the current study design we cannot confirm whether this is the case or not.

We saw similar patterns in males and females regarding the effect of maturation on the assessed immune markers. Although males showed somewhat lower capacity for bacterial growth inhibition compared to females, this difference was only marginally significant, and there was no sex difference in cell wall lytic capacity. While gamete production is expected to be more costly in females, our observations do not show stronger effect of maturation in females in either of the immune measures. In the past, costs of sperm production in males was thought to be relatively 'cheap', but recent studies show that males also pay considerable energetic costs for gamete production and storage (*Michalik & Rittschof, 2011*), which costs might have contributed to similar energetic conflicts between the tested immune markers and reproductive investment during maturation in the sexes. Notably, due to the low final sample sizes of data on the laboratory reared spiders drawing general conclusions was made with caution, as some ambiguities in our results may have resulted from the low sample size, such as the absence of significant sex-differences in the immune measures, or their association with life stage. Nevertheless, the similar decrease in lytic activity in both sexes might suggest that not only gamete production, but maturation itself also has considerable energetic costs that might mask sex-differences in the changes in immunity associated with the life stage transition.

While mated females collected in late summer showed a much more prominent decrease in cell wall lytic activity than in bacterial growth inhibition, and subadults and adults only differed in cell wall lytic activity, we found no significant linear association between cell wall lysis and fecundity. It should be noted, however, that a considerable number of samples from mated females failed to induce measurable clearing zones on the cell wall agar. Consequently, the number of observations in which we could regress fecundity on non-zero values of cell wall lytic activity was quite low, rendering it difficult to identify a credible linear association between these variables. Based on the binomial models, females with higher fecundity were more likely to have undetectable lytic activity, although this observation was significant only in late summer females. Bacterial growth inhibition showed no stage-differences, and was only slightly decreased in mated females. Note, however, that bacterial growth inhibition showed considerable negative correlation with fecundity, both in spring and late summer females. While this correlation was not significant in the case of lytic activity, more fecund late summer females were still more

likely to exhibit zero cell wall lytic competence. Altogether, these observations seem to support that physiological costs of production and storage of the wide variety of antimicrobial biomolecules may interfere with elements of reproduction. Indeed, reproduction (and reproductive effort) is often shown to be negatively associated with elements of immunocompetence (*Nordling et al., 1998*; *Schwenke, Lazzaro & Wolfner, 2016*).

We propose that lower adult capacity of bacterial cell wall lysis, and decreased capacity to inhibit bacterial growth and to break down cell wall in mated females characterised by high fecundity, may be due to that adults have only one short time period for reproduction. As such, the trade-off between immunity and fecundity might be expected to be quite substantial in species that reproduce only once throughout their lifetime. The main argument for this is that they have to invest as much as possible into the first reproduction in order to maximise reproductive output, because in semelparous organisms there are no future reproductions, i.e. the first reproduction is also the last. As such, this may lead to a 'quasi-terminal investment' into reproduction, in which an individual might take the risk of having some elements of immunocompetence at a decreased efficiency duo to decreased investment into somatic maintenance, for the prospect of increased reproductive success. The negative correlation between bacterial growth inhibition power and fecundity, and the decreased capacity of producing measurable bacterial growth inhibition and cell wall lysis in more fecund females appear to support this hypothesis, as those females that 'gave up' more capacity in the assessed immune measures tended to have a larger number of offspring. Maximising reproductive output through a terminal investment-like scenario would be expected to be favoured by natural selection in environments where adult future mortality rate is high (assuming that this high mortality is not due to pathogens), which is known to be characteristic to populations of semelparous species (*Goldstein, Merrick & Koprowski, 2017*; *Sæther, 1988*; *Young, 2010*).

Differences in expected pathogen diversity or prevalence were also proposed to drive the evolution of life stage differences in immune function (*Wilson-Rich, Dres & Starks, 2008*; *Shi & Sun, 2010*). We, however, find this proposition unlikely to be the main cause of the difference in lytic activity between subadults and adults in *P. agrestis*, because they reside in the same habitat and are quite the same in their foraging habits and preys (*Nyffeler & Benz, 1988*; *Samu & Szinetár, 2002*; Z. Rádai, 2016, personal observations).

Because both the maintenance of an efficient immune system and a higher rate of post-embryonic development are considered to be costly, one might expect the presence of a negative correlation between them (*Modak et al., 2009*; but see *Rantala & Roff, 2006*). However, we found no significant trend regarding the relationship between growth rate and bacterial growth inhibition or cell wall lytic activity in the laboratory-reared spiders. Notably, ad libitum availability of food might have mitigated the energetic conflict between these traits leading to the observed results (*Brzek & Konarzewski, 2007*; *Mangel & Stamps, 2001*). In a previous study we have found a positive relationship between age at maturation and encapsulation efficiency in *P. agrestis* adults (*Rádai, Németh & Barta,*

*2018*), meaning that individuals characterised by faster development showed weaker encapsulation response. In the cricket *Gryllus bimaculatus*, developmental time was found to be positively correlated with lytic activity of the hemolymph, but was negatively associated with encapsulation rate (*Rantala & Roff, 2005*).

The result that no significant relationship was found between bacterial growth inhibition power and cell wall lytic activity either suggests that there is little, or no energetic conflict between these two immune parameters, or that other, in physiological sense costly, traits obscure this conflict between the two assessed elements of immunity. A potential energetic conflict may also have been masked by the unrestricted food availability of the laboratory reared specimens. Also, the absence of a positive association implies that individuals that invest relatively more in bacterial growth inhibition are not necessarily able (or prone to) invest similarly high amount of resources into lysozyme production (and vice versa). It should be noted, though, that while cell wall lytic activity is mainly dependent on lysozymes, bacterial growth inhibition results from the presence of many types of peptides and proteins, which lysozyme is also a part of. Hence, the failure to detect a significant relationship between bacterial growth inhibition and cell wall lytic activity may be due to the sheer number of biomolecules with a role in bacterial growth inhibition that can have diverse associations with lysozymes, rendering it difficult to identify a consistent association between them. It is also noteworthy that in the case of other elements of immunity it is often found that there is no strict association between assessed immune parameters (*Fedorka, Zuk & Mousseau, 2004*; *Gershman et al., 2010*; *Kortet, Rantala & Hedrick, 2007*; *Piñera et al., 2013*).

## CONCLUSIONS

Overall, our results show that in a semelparous species adults are characterised by lower cell wall lytic activity in comparison with subadults, but we found no evidence for subadults and adults to differ in their bacterial growth inhibition power. We argue that a likely explanation for relatively higher subadult lytic activity may be associated with an energetic conflict leading to a shift in prioritisation between somatic and reproductive tissues in adults. This conflict is expected on the basis that adults of a semelparous species don't face the trade-off between current and future reproduction therefore maximal reproductive success may be achieved by allocating more resources into reproductive tissues, rather than in somatic maintenance and long-term survival. Also, bacterial growth inhibition was negatively associated with fecundity in mated females, hinting at the presence of a trade-off between investments into reproduction vs. somatic maintenance. Indeed, among late summer specimens mated females had much weaker bacterial growth inhibition power and cell wall lytic activity than virgins. Admittedly, in future studies higher sample sizes and better balanced study designs will be needed in order to clarify the remaining ambiguities of our results. Also, comparative studies on both semelparous and iteroparous species would undoubtedly help to reconcile the role of reproductive strategies in shaping trade-offs between key life history traits. We argue that *P. agrestis* would be an excellent model organism to study both the intrinsic and extrinsic factors that might contribute to these elements of life history evolution.

# ACKNOWLEDGEMENTS

We are indebted to Dr. Tamás Emri for his aid in, and providing laboratory space for, the conducted measurements, and to Dr. Attila Bácsi for his aid in long-term storage of our samples. Also, we thank Dr. Ádám Kiss for his tools and help in photographing the spiders. We are grateful to Dr. Jácint Tökölyi and three anonymous reviewers for their constructive comments on the manuscript.

## Funding

Zoltán Barta was financed by the Higher Education Institutional Excellence Programme of the Ministry of Human Capacities in Hungary, within the framework of the FIK-Lendület Behavioural Ecology Research Group thematic programme of the University of Debrecen. The work/publication was also supported by the EFOP-3.6.1-16-2016-00022 project. The funders had no role in study design, data collection and analysis, decision to publish, or preparation of the manuscript.

## Grant Disclosures

The following grant information was disclosed by the authors:
Higher Education Institutional Excellence Programme of the Ministry of Human Capacities in Hungary, within the framework of the FIK-Lendület Behavioural Ecology Research Group thematic programme of the University of Debrecen.

## Competing Interests

The authors declare that they have no competing interests.

## Author Contributions

- Zoltán Rádai conceived and designed the experiments, performed the experiments, analysed the data, prepared figures and/or tables, authored or reviewed drafts of the paper, approved the final draft.
- Péter Kiss performed the experiments, authored or reviewed drafts of the paper, approved the final draft.
- Dávid Nagy performed the experiments, authored or reviewed drafts of the paper, approved the final draft.
- Zoltán Barta contributed reagents/materials/analysis tools, authored or reviewed drafts of the paper, approved the final draft.

## Data Availability

Raw data are available in the Supplemental Files.

## Supplemental Information

Supplemental information for this article can be found online at http://dx.doi.org/10.7717/peerj.7475#supplemental-information.

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
