# Peer review of "Antibacterial immune functions of subadults and adults in a semelparous spider"

_PeerJ, doi:10.7717/peerj.7475_

## Round 0.1 · original submission · Major Revisions

Dear Dr. Rádai and colleagues:

Thanks for submitting your manuscript to PeerJ. I have now received three independent reviews of your work, and as you will see, the reviewers raised some concerns about the research. Despite this, these reviewers are optimistic about your work and the potential impact it will lend to research on spider antimicrobial immune responses. Thus, I encourage you to revise your manuscript accordingly, taking into account all of the concerns raised by the reviewers.

Please pay careful attention to the concerns raised regarding your statistical analyses. Also, in your Discussion, please discuss in more detail the knowledge gap that is addressed by your findings (focus on spiders and then compare with what is known for related arthropod groups).

I look forward to seeing your revision, and thanks again for submitting your work to PeerJ.

Good luck with your revision,

-joe

Reviewer 1 ·

Basic reporting

Please see the general comment section

Experimental design

Please see the general comment section

Validity of the findings

Please see the general comment section

Additional comments

Rádal et al. present a comparison of two immunity functions tests in subadults and adults in a spider species. They also compare those tests between virgin and mated females from the wild. The central question is well-stated (the difference in investment for semelparous species particularly well introduced), the data seem sound (even if their limits need be better acknowledged), and the methods are well-presented. Nevertheless, I believe there are a couple of key points that need addressing to insure the validity of the findings and conclusions stated here.

- The authors base their hypothesis on the fact that ‘the production of gametes in sufficient amount (and quality) [is done] shortly after maturation’ (l.66-67). Because they use subadult and adult stages, a key biological factor is that gametes (and other reproductive tissues) are only produced during the adult stages, in males and in females, in this species. I don’t see this to be supported here (it may be very well-known in spiders, but the information needs to be there). If this is not the case, I believe the study is still interesting, but the argument about somatic/ reproductive investment should only be mentioned as a potential explanatory factor rather than a central point of the argument (and this should be discussed). In this regard, it would be interesting to know more about the mean duration of the adult/ subadult stages in this species. In particular, if the adult stage is very short, it’s very likely that the difference in somatic/ reproductive investment starts before the adult stage.

- It’d be good to understand the physiological differences between the two assays, from the introduction. The only little information are at the very end of the MS (l.458-460), and not sufficient still. Is the lysozyme assay more ‘generalist’ than the antimicrobial peptide one? Would the same results be expected with other bacteria than the ones used in both assays?

- Statistical analyses 1: the MCMCglmm model: responses are a ratio between immunity assays and cube of prosoma width, and growth rate, a predictor in the model, is a ratio between cube of prosoma width and developmental time. Using these ratios does makes sense biologically, but it doesn’t in terms of the model, that has the same variable (cube of prosoma width) on the sides of the = sign

- Statistical analyses 2: repeatability measures: ratios between variance per individual and total variance are used . The problem is that repeated measures for an individual were always performed in the same Petri dish. So, individuals and Petri dishes are completely confounded in these measures. Hence there isn’t any meaning in measuring the repeatability of individuals, that could be completely due to variance between Petri dishes...

More minor comments:
- The limits of the sample size need to be clearly acknowledged in the discussion. There are only 4 adult males used the first experiment… So, it is quite speculative to draw much conclusion about the lack of relationship found in this experiment (and I believe the discussion could then also gain from a little shortening following this)
- Line 124-126: something is wrong with the numbers 154≠40+68+23+28
- Line 202: I don’t see why the authors divided all clearing zone area by the area of the petri dish if all petri dishes are the same size… Is it a problem in the wording? Dividing all assay results by a constant will have absolutely no incidence in the analyses, it will even not be visible when results are later z-scored…
- L. 290: Why is it 90% and not 95%?
- Line 387-388: maybe a test of with/ without inclusion of the induction of clearing zone would be of interest? (really as a potential suggestion here)
- Line 397-399: didn’t understand that sentence…
- There are a few English mistakes here and there:
- l.22 (don’t)
- l.52: cycle (singular)
- l. 116 ‘in’ should be removed
- l.277 “being saId”

·

Basic reporting

1. BASIC REPORTING
a. Clear, unambiguous, professional English language used throughout.
i. The language throughout the manuscript requires improvement and would benefit from the following suggested revisions:
1. Line 15: Phrase “were proposed” is used. Proposed by whom? By the authors, or someone else? If the latter, this needs a citation. If the former, the sentence should read “We propose” or “We proposed.” Unless this supposed to read “are proposed” as in, in general, this is what is commonly proposed in the literature. This would also require a citation.
2. Line 17: Change “stage-level” to “stage-specific.”
3. Line 17: Is the phrase “juveniles and subadults” redundant, since these are synonyms? If so, I would remove “juveniles” since subadults is used throughout.
4. Lines 17-19: Revise wording. For example, phrases like “juveniles and subadults are yet unable to reproduce” could be stated more clearly by removing “yet,” while the word “certainly” could be removed from line 19.
5. Lines 19-22: This sentence is saying several things and should be split into at least two sentences.
6. Lines 61-63: Add the word “and” before “hence cannot trade off…”
7. Line 81: Revise the language of the phrase “assay is based on that…” to read something like “assay is based on the fact that…”
8. Line 92-93: The hemocoel encompasses the entire open body cavity, yet this sentence makes it seem as if it is restricted only to the abdomen. Is that really the case? Wouldn’t the hemocoel also extend into the cephalothorax in this spider? Or is that a separate body cavity?
9. Line 103: Revise the phrase “with the used immune parameters” to something like “with the immune parameters we used” or “with the immune parameters we utilized.”
10. Line 106: Use the word “Henceforth” rather than “Henceforward”?
11. Line 106: Add “refer to” after the phrase “we use the term ‘subadult’ to” and add “to” after the phrase “one moult prior…”
12. Line 108: Remove the word “have” after “We” at the start of the sentence. Similar corrections can be made throughout this section and elsewhere.
13. Line 114: The phrase “ad libitum” should be italicized. Further, as it is used here the phrase describes a property of the water and food, when it should be a descriptor of the spiderlings’ feeding behavior (i.e., “the spiderlings fed on … ad libitum”). Please modify.
14. Line 116: Remove the word “in” before the phrase “every two days…”
15. Line 122: Remove the comma after “collected subadult…”
16. Line 123: Remove the comma after “in 2017…”
17. Line 129: The word “latesummer” should be “late summer.”
18. Line 136: The phrase “killed in” should be “killed at” unless you specify the manner in which they were killed at -20˚C.
19. Line 137: Remove “a” from the phrase “because a the heart…”
20. Line 144: Remove the comma after “ice…”
21. Line 145: Should say “at 4˚C” not “in 4˚C…”
22. Line 146: “20-20 μl” a typo? If so, please correct.
23. Lines 165-168: Revise this sentence, as the current wording is difficult to follow.
24. Line 277: The phrase “being sad” should be “being said…”
25. In general, the “Developmental stage differences in immune markers” and “Association between reproductive status, fecundity and immune markers” subsections of the materials and methods section could be more concisely.
26. Line 284: Add “a” before the phrase “Chi-squared test…”
27. Line 401: Revise “quasi showing a” phrase to “showing a quasi…”
28. Line 473: Add “the fact” before the phrase “that adults…”
29. Line 474: Remove the comma after “future reproduction” and add “and” in its place. Also, change “maximizing” to “maximal…”
b. Intro & background to show context. Literature well referenced & relevant.
i. Lines 72-78: The text introduces the study organism but does not provide its common name or other appropriate details about its life history (although some of this is mentioned in the abstract). So far, the reader knows that it is a semelparous species with adult and subadult stages, but a sentence or two here explaining the life history in greater detail would be helpful.
ii. The introduction says much about immune trade-offs in adults due to investment in somatic maintenance and reproduction, but little is said of how trade-offs with reproduction and fecundity in females might differ with similar trade-offs observed in males. Consequently, the rationale for assessing fitness in both virgin and mated females as well as in males is not explicitly delineated in the text. Please explain the rationale for looking into both sexes (as well as both virgin and mated females) and how you might expect adult immunity to differ (if at all) between them.
c. Figures are relevant, high quality, well labelled & described.
i. Fig. 1: Panel (b) subadult and panel (d) adult box plots do not show the minus SD error bars. Level of significance for a single asterisk should also be mentioned in figure legend.
ii. Fig. 2: Panel (a) and (c) box plots are missing what I assume to be minus SD error bars (the legend should specify what the error bars represent). Level of significance for a single asterisk should also be mentioned in figure legend. The panel (d) mated box plot appears to be missing and shows what appear to be individual activities (circles) unlike the other graphs. Please correct for consistency with the other panels.
iii. Fig. 2 panel (a) shows a significant difference between virgin and mated spring females, but the box plots look nearly identical. However, the text states that “mated females showed significantly lower levels of bacterial growth inhibition (Fig. 2a-b, Table 2)” and Table 2 has a p value of P=0.030. Please double check that this is accurate and that the statistical test used is both appropriate and not yielding a false positive here.

Experimental design

2. EXPERIMENTAL DESIGN
a. Research question well defined, relevant & meaningful. It is stated how the research fills an identified knowledge gap.
i. Line 106: Please explain the rationale for why penultimate subadults were chosen instead of earlier subadults.
b. Rigorous investigation performed to a high technical & ethical standard.
i. Can you further explain the rationale behind choosing two assays, a bacterial growth inhibition assay and bacterial cell wall lytic activity assay, that measure partially overlapping humoral immune parameters? As you stated earlier in the introduction with respect to the bacterial growth inhibition assay “These molecules [AMPs] mainly attack the integrity of the cell wall and cell membrane of the invading microbes.” However, as you note in the discussion (lines 458-460), lysozyme activity is also at play here. Likewise, the cell wall lytic activity assay may also be partially a result of AMP lytic activity, and not lysozymes alone, so you may need to qualify this somewhere in the text as well. In short, I believe your manuscript would be improved by providing your readers a stronger justification for using of both assays here and how they provide different information despite similar modes of action and partially overlapping activity. Distinguishing the two further should only require a few additional sentences in your introduction and/or discussion sections.

Validity of the findings

3. VALIDITY OF THE FINDINGS
a. Data is robust, statistically sound, & controlled.
i. Lines 119-121: This is a rather low sample size (n=38 overall) to subdivide into 4 groups. There were only 4 male adults from the 2016 collection, for example. Please explain how this sample size was sufficient given the variation within your groups.
ii. At first glance, it appears somewhat concerning that of the 68 mated spring females, only 25 cocoons yielded fecundity data, whereas 28/28 of the mated late summer female cocoons yielded data. Was embryo viability and parasitism much higher in the spring compared to the late summer? If so, how would this potentially confounding factor be accounted for in your analyses, particularly those that compare data across seasons (for example, see the comparison made in lines 304-306)?
iii. Lines 289-291: The sentence on the Bayesian analysis seems somewhat out of place here and its biological significance is not made clear to the readers.
iv. Lines 296-299: Again, the biological significance of these statistical tests needs to be briefly stated here.
v. Lines 314-319: Please explain the biological significance of these statistical measures.
b. Conclusions are well stated, linked to original research question & limited to supporting results.
i. Lines 340-344 state “On the other hand, mature individuals have to invest resources into gonad development and gamete production. Consequently, a higher proportion of the acquired energy and nutrients can be assigned to somatic tissues and to elements of somatic maintenance in subadults, while these limited resources will be partitioned into reproductive tissues and gamete production as well in adults.” Were the adults used in these experiments all virgins? If so, were the cell wall lytic activity results surprising, or would you expect lower immunity in adult females regardless of mating status? Also, when do females produce cocoons? Would you expect immune trade offs before, or only after process? And, in males, you saw similar declines in lytic activity as observed in females, but presumably males are investing fewer resources than females to gonad/gamete production. Can you explain why you saw similar trends in both sexes with the cell wall lytic activity assay? Apologies if I overlooked this.
ii. Line 348: Could the lack of bacterial growth inhibition changes observed here be due to the bacterial strain used? This may explain why the lytic assay, which involved a different bacterium, did show ontogenetic differences.
iii. Line 352: You may need a citation for the pollen beetle work here too and not just after the following sentence on line 356.

Reviewer 3 ·

Basic reporting

The manuscript meets the standards:
-The manuscript is written in clear and unambiguous, professional English.
- The manuscript present correct article structure, figures, tables.
- The manuscript is self-contained with relevant results to hypotheses.

I propose changes in the introduction to cover the field of knowledge in a not so general way, to express the main theories differently and rethink the basal hypotheses of the work.

Experimental design

The manuscript meets the standards:
-The research is within the 'Aims and Scope' of the journal.
- The research question is well defined, relevant & meaningful.
- The manuscript has been conducted rigorously and to a high technical standard, and in conformity with the prevailing ethical standards in the field.
- The methods described in the manuscript are sufficient detail & information to replicate.

It is not very well expressed how research will fill an identified knowledge gap, and there is little information about the taxonomic group evaluated as well as citations in related taxonomic groups.

The methodology of the manuscript is very thorough, the statistical analysis has been performed rigorously, and the experimental design is correct. I only suggest taking into account some considerations for the statistical model that evaluates fecundity in function of antibacterial immune functions.

Validity of the findings

The manuscript meets the standards:
-The data is robust, statistically sound, and controlled.

Also, although the manuscript gives good and stimulating reading, the discussion of the results obtained and the conclusions drawn by the authors must be re-examined in a new version of the manuscript.

Additional comments

Detailed remarks (numbers refer to lines in the manuscript):

37: ‘Immunocompetence is a critically important element of somatic maintenance’ Why? How? I think a broad definition of immunocompetence is missing.
48: Add: - Rachel Gilbert, Richard D. Karp, George W. Uetz, Effects of juvenile infection on adult immunity and secondary sexual characters in a wolf spider, Behavioral Ecology, Volume 27, Issue 3, May-June 2016, Pages 946-954 , [url]Https://doi.org/10.1093/beheco/arv241[url]
63-64: It's a rule? I believe that this approach is somewhat reductionist, and will depend on multiple factors (ecological, sexual, time of life of the species, mating system, sperm production, parental care, etc.). There is no citation on this topic in all this paragraph, it would be good to mention literature where this trade-off has been seen. The discussion on the differences in allocation of resources (somatic vs. reproductive) between species semelparous vs. Iteroparous is fine, but it can not be a central point or central hypothesis of the work since you are not comparing species with these reproductive characteristics. I think it is merely speculative to attribute the differences in immunological parameters observed between stages to this. I think it can be discussed and proposed speculatively, and perhaps discuss possible future comparative works among species of the genus that possess other reproductive tactics.
70: Will adults invest less in immunocompetence than juveniles? How would that be? Immunocompetence is the resource trade-off that individuals face between reproduction and other activities (as immunological traits), but in juvenils there would be no trade-off because there is no reproductive investment. Perhaps adults invest less in the production and maintenance of immunological parameters than in juveniles, but not if it is correct to talk about investment in differentiated immunocompetence between stages. Please review this concept throughout the entire work.
72: Why is Pardosa agrestis a good model for this study? What is known about this species that may be of interest for this study? I see that the Ph.D. thesis of Kiss, 2003 is cited but no other work.
75-76: ‘This semelparous spider species has, by definition, only one, short and well defined reproductive event during its lifetime’ What does that mean? Is it a monogamous species? Or is it that these species only have one reproductive season? Is anything known about the sperm production of this species or closely related species?.
83: Add a citation.
98-101: Sentence with repetitions, rewrite please.
111-112: Please provide the geographic coordinates for the collection area. The authors collected spiders in two nearby habitats, and this variable was not included in the models as a random effect, are the environments sufficiently similar? Could there be differences at the microhabitat level? Drapela et al., 2011 found that the environment can influence several reproductive characteristics of P. agrestis. I think it would be interesting to take this into account.
Drapela, T., Frank, T., Heer, X., Moser, D., & Zaller, J. G. (2011). Landscape structure affects activity density, body size and fecundity of Pardosa wolf spiders (Araneae: Lycosidae) in winter oilseed rape. European Journal of Entomology, 108(4).
122: The second collection was in the spring of 2017 and the third collection in the summer of 2017? What does this mean biologically for the species? Here I miss some basic information if it exists about when is the reproductive season, or about the lifetime duration of two sexes?
127: Out of topic but interesting to keep in mind: could you see what characteristics the parasitized cocoons had: were they smaller? Were they less heavy? Did they have any special characteristics? Then see the females that produced these cocoons, if they had lower body condition for example? And finally, correlate the immunological parameters of the females with the occurrence of parasitism. Perhaps females with lower immunological values are more likely to be parasitized. It would also be interesting to see if there are differences between seasons or habitats in the occurrence of parasitism.
146: ‘20-20 μl from the supernatant’, clarify the quantity.
152: Add a citation with previous information about these periods of time have no consequences on the antimicrobial activity that will be measured.
243: Is there another work that has previously used this approach?.
256: The variable ‘season of collection’ was added as a variable confounding in this case? (as in the next analysis)? Or did you have any particular prediction or interest in the differences between seasons?.
257: The habitat variable (collection site) was not included in the models as a random effect, are the environments sufficiently similar? Could there be differences at the microhabitat level?.
264: I am not very familiar with Bayesian statistics although I see that there could be some problems with this model. The fecundity variable is represented by the number of offspring of each mother (only named at the end of the introduction) and is probably a non-parametric variable. Is there any Bayesian approach for data of this type? (e.g., Müller & Quintana, 2004; Hjort et al., 2010). Did you check the collinearity of the predictor variables (bacterial growth inhibition and cell wall lytic activity)? In the second model, it was found that the season had an effect on the cell wall lytic activity and the bacterial growth inhibition (304, 307-308). Now you are setting as predictor variables: 'season' and the antibacterial immune functions with which previously season had been related. There is a high risk of collinearity between the predictor variables, it is necessary to check this previously (perhaps controlling the variance Inflation factor). In the discussion, it is expressed that: ‘the number of observations in which we could regress fecundity on non-zero values ​​of cell wall lytic activity was quite low’, which can also be clearly seen in figure 3b. Is there no way to deal with data that has many zeros? (something analogous to zero-inflated models) or make a model without the non-zero observations. Finally, according to Drapela et al., 2011 clutch size was positively correlated with female body size in P. agrestis. Do not you think it is necessary to control the number of offspring by the size of the female or add the condition as a predictor variable?.
Müller, P., & Quintana, F. A. (2004). Nonparametric Bayesian data analysis. Statistical science, 95-110.
Hjort, N.L., Holmes, C., Müller, P., & Walker, S.G. (Eds.). (2010). Bayesian nonparametrics (Vol. 28). Cambridge University Press.
276: ‘That being sad we dediced ..’ Rewrite.
335: See review of lines 63-64. The fact that juveniles invest more in somatic maintenance vs. adults could be because they should not allocate resources in reproduction as adults. However, attributing the differences found between age stages to which the species is semelparous is not possible with the experimental design proposed (you should compare adults vs. juveniles in species with the two reproductive tactics).
340: "On the other hand ..." Remove.
340-344: Sentence with repetitions, rewrite please.
347: Male and female gametes have different costs, so how do you explain that no differences were found in the immunological parameters between sexes?
351: Too much information is given about the screened APTs in the work of Vogel et al., 2014. Compare in a more concise and meaningful way.
360: The experimental design is not to conclude on the costs of producing/storing lysozymes, please rewrite this phrase in a more speculative way. The example of Vogel et al., 2014 is not correct for this part of the discussion (they do not find an 'energetic conflict' but differences in the composition of lysozymes between stages).
374-381: The explanation of the differences found between the seasons is not clear to me. What does it mean that the females of these seasons belong to two separate generation cohorts? There is lacking basic information of the species.
Check: Samu, F., Németh, J., Tóth, F., Szita, É., Kiss, B., & Szinetár, C. (1998). Are two cohorts responsible for bimodal life history pattern in the wolf spider Pardosa agrestis in Hungary. In Proceedings 17th European Colloquium of Arachnology (pp. 215-221).
381: ‘segment of the spider's life history’ Rewrite
388-390 See review of line 264.
396: Add more recent literature.
397: See review of lines 63-64, 355. Please note that the experimental design does not support these ideas. Then I have a problem in the approach that the authors give to the ‘terminal investment’. In the work of Creighton et al., 2009 there is no mention of reproductive strategies of species semelparous vs. Iteroparous, but of the strategy of each individual for multiple matings and trade-offs with future matings (i.e., the shift in allocation towards current reproduction with increasing age). Iteroparous or semelparous species can make a terminal investment if they invest more resources and energy in their last mating compared to the previous ones, it is an intra-individual approach that has to do with the age of the individual and the senescence. Please review the concept and better explain its use for this case.
Check: Velando, A., H. Drummond, and R. Torres. 2006. Senescent birds redouble reproductive effort when ill: confirmation of the terminal investment hypothesis. Proceedings of the Royal Society B: Biological Sciences 273: 1443-1448
Bonneaud, C., Mazuc, J., Chastel, O., Westerdahl, H., & Sorci, G. (2004). Terminal investment induced by immune challenge and fitness traits associated with major histocompatibility complex in the house sparrow. Evolution, 58(12), 2823-2830.
Charlesworth, B., and J. A. Leon. 1976. The relation of reproductive effort with age. American Naturalist 110: 449-459.
Clutton-Brock, T. H. 1984. Reproductive effort and terminal investment in iteroparous animals. American Naturalist 123: 212-229.
412. Sentence with repetitions, rewrite please.
414: Add citations.
423: ‘Furthermore, in the species...’ change to ‘furthermore, in several species’.
Figure 2: Put n in capital letter to match with Figure 1. I am struck by the graph (d), why is the data structure like this for mated females in late summer? Do you have many similar data? Please explain.
Figure 3: Put n in capital letter to match with Figure 1. See review of line 264.
Table 1: ‘For discrete predictors’, maybe better ‘for categorical predictors’.
Tables: I suggest unifying the tables (at least 1 and 2) into one.

---

## Round 0.2 · Minor Revisions

Dear Dr. Rádai and colleagues:

Thanks for revising your manuscript. The reviewers are very satisfied with your revision (as am I). Great! However, there are several remaining concerns by both reviewers (mostly minor and/or grammatical). Please address these ASAP so we may move towards acceptance of your work.

Best,

-joe

·

Basic reporting

i. Lines 20-23: Sentence was split in two in an awkward manner. Please revise.
ii. Line 67-68: Remove phrase “to see.”
iii. Line 70: Remove comma before “because.”
iv. Line 71: Add “a” before “critical impact.”
v. Line 83-84: Consider placing the text “i.e. there are two separate adult peaks throughout the year” in parentheses.
vi. The paragraph in lines 122-132 is a welcomed addition, but seems out of place in an introduction, as it now goes into far more detail on the specifics of the two immune assays than before. Either re-locate to the “Materials and methods” section or significantly shorten (or both).
vii. Lines 125-126: Consider placing the text “i.e. are expected to be effective against microbes in general when released simultaneously” in parentheses.
viii. Line 134: Delete “in the bacterial growth inhibition power and cell wall lytic activity, i.e.” If you’d like, you can add parenthetical references to each immune response mentioned in lines 135-136.
ix. Line 215: Change “persistm" to “persist.”
x. Line 448: Replace “Albeit” with “Although.”
xi. Line 480: Add “a” before “stronger effect.”
xii. Line 511: Add “the fact” before “that adults.”
xiii. Line 515: Change phrase “no future reproductions” to “no future opportunities to reproduce” and consider removing the phrase “i.e. the first reproduction is also the last” as this phrase is redundant and the “i.e. …” pattern is overused in the revised portions of the text.
xiv. Line 517: Change “duo” to “due.”
xv. Line 531: Change “the same” to “similar.”
xvi. Line 547: Change “, in physiological sense costly,” to “physiologically costly.”
xvii. Line 554: Change “bacterial growth inhibition results from the presence of many types of peptides and proteins, which lysozyme is also a part of” to “bacterial growth inhibition also results from the presence of many other types of peptides and proteins.”
xviii. Line 702: Change the informal “don’t” to “do not.”
xix. Line 703: Add “and” before “therefore maximal.”

Experimental design

no comment

Validity of the findings

no comment

Additional comments

Most of the suggested edits were taken into account in the revised manuscript. When suggestions were not followed, the authors provided sufficient rationale for not doing so or for doing so in a modified manner. Clarifications on questions from the first review were satisfactory. The revised portions of the text generally read very nicely. I have provided a number of minor suggestions for revisions to improve the language in some of the revised portions of the document. If these additional changes to the text are made I recommend that the manuscript be accepted without further review.

Reviewer 3 ·

Basic reporting

The manuscript meets the standards:
-The manuscript is written in clear and unambiguous, professional English.
- The manuscript present correct article structure, figures, tables. The raw data is shared correctly.
- The manuscript is self-contained with relevant results to hypotheses.

I suggest covering in a better way and more completely some fields of knowledge in the intoduction (sexual dimorphism in immune response or mating relationship with immunity), to formulate more precise predictions regarding these topics.

Experimental design

The manuscript meets the standards:
-The research is within the 'Aims and Scope' of the journal.
- The research question is well defined, relevant & meaningful.
- The manuscript has been conducted rigorously and to a high technical standard, and in conformity with the prevailing ethical standards in the field.
- The methods described in the manuscript are sufficient detail & information to replicate.

The methodology of the manuscript is comprehensive, the statistical analysis has been performed rigorously, and the experimental design is correct.

Validity of the findings

The manuscript meets the standards:
-The data is robust, statistically sound, and controlled.

Conclusions are well stated, linked to original research question. However, several paragraphs should be re-ordered and re-examined in a new version of the manuscript. The addition of citations is requested in various parts of the discussion.

Additional comments

*The lines correspond to the final revised document (without tracked changes).

General comment:
It seems to me that the authors have correctly introduced, predicted and tested that individuals manage their resources in a stage-specific manner (subadults invest more in immune function and adults allocate their resources between reproduction and immune function). Probably this general phenomenon occurs in both semelparous and iteroparous species. It is possible, as the authors suggest that this trade-off between activities across ontogenetic stages is more pronounced in a semelparous species since only have one reproductive season. However, the authors are based on the fact that the species is semelparous to formulate its hypothesis and explain its results, which is incorrect since no specific design was made to test this (and it is excellent that the authors suggest it in the conclusions). Therefore, the authors should be even more careful with the approach of semelparity as an explanation of the patterns found. And in the excessive importance given to this strategy throughout the manuscript: TITLE, Abstract (21-25), Introduction (63-73), Discussion (394-397, 489-507), Conclusion (548-552).

Detailed remarks (numbers refer to lines in the manuscript):

Introduction:
67: "to be pronounced in semelparous species". Pronounced about what? The comparison is missing.
69: "no future reproductions". Better “no future reproductive seasons”.
73: "somatic maintenance" rewrite to avoid repetition.
62-76: This paragraph has some repetitions with the previous one, and has no citation.
78: "two separate adult peaks". Maybe better: two separate phenological peaks?
81-82: "sperm is then excreted and stored". I do not think that "excreted" is the best term, are you talking about sperm induction? Rewrite.
80-84: If there is no work to be cited on life history here put as personal observations of the authors.
88: "one would expect…" Better "we expect".
89: Unify concepts (e.g. 59: juvenile and subadults; 73: preadult; 89: premature).
90: "For proxies of immunocompetence…". I think the concept of immunocompetence was finally changed, since it can only be measured in adults. I suggest removing the term and writing better "For proxies of immune function" or something like that. If the term "immunocompetence" is maintained, it should be clarified that it will measure in adults (e.g. comparing virgins vs. mated, or males vs. females) and the concept must be previously defined.
110: "production of bacteria". Maybe "reproduction of bacteria?"
123-126: I understand that the axis of the manuscript is the comparison between adults and subadults. However, I notice that the introduction is extremely unbalanced in the presentation of the topics. Only in these lines the authors present some backround about the sexual dimorphism in immune response and there is no prediction about this topic. In some statistic models, sex was included as a predictor variable and in the discussion, there is a fairly extensive paragraph discussing the results. I suggest that the authors should introduce more extensively this topic. 126-130: Something similar to the previous comment. Only in these lines is the relationship between the immune response and fecundity. It seems to me that the theoretical framework on these points should be expanded, since the models include the reproductive state, there is a model with the fecundity as response variable, and another where the measurability of immune function is related to fecundity, and then in the discussion this topic is addressed in an extensive way.

Materials and methods:
197: Order the citations chronologically.
200: "persistm" correct typo.
352-353, 378-379: The interactions are not significant, but are they removed of the final models? The coefficients and estimates given are of the model without including the interactions?. If the interactions are excluded from the models, how are they excluded, with some particular method?
371-374: "the interaction terms were excluded from the final model". How is it excluded, with some particular method?
360-365: the values of Coefficients and PMCMC of the contrasts between virgin and mated females should coincide with the contrasts presented in table 2? Why do not they coincide?

Discussion:
410: "In M. aeneus it was…". As it is the first time that this species is named, put the scientific name without abbreviations.
414-416: Move the citation "(Gollas-Galvan et al., 2017)" at the end of the sentence.
416,446,451,464,518: the authors repeat the word "Notably", try to replace it sometimes with synonyms.
431: Sometimes a species is presented in brackets, sometimes not, sometimes with its vulgar name, sometimes not. Unify this in the examples.
438: "the cricket Gryllus texenis males", perhaps "adult males"?
429-454: I see the discussion somewhat messy. I suggest moving these paragraphs that talk about differences in immune function between subadults and adults to line 416 before you start talking about the relationship of immunity with fecundity.
462: Add more general citations
e.g., Dewsbury, D. A. (1982). Ejaculate cost and male choice. The American Naturalist, 119
(5), 601-610.
Wedell, N., Gage, M.J., & Parker, G.A. (2002). Sperm competition, male prudence, and
sperm-limited females. Trends in Ecology & Evolution, 17 (7), 313-320.
455-470: It's ok the discussion about the lack of differences between sexes, but I think you need to add citations of works with similar results
e.g., Giglio et al., 2016 (citation again)
Meylaers, K., Freitak, D., & Schoofs, L. (2007). Immunocompetence of Galleria mellonella:
Sex- and stage-specific differences and the physiological cost of mounting an immune
response during metamorphosis. Journal of Insect Physiology, 53 (2), 146-156.
Yourth, C. P., Forbes, M. R., & Baker, R. L. (2002). Sex differences in melanotic encapsulation
responses (immunocompetence) in the damselfly Lestes forcipatus Rambur. Canadian
Journal of Zoology, 80 (9), 1578-1583.).
Something important and that was not mentioned is that the sexual comparisons were made with virgin individuals, who had not faced another reproductive cost than the production of gametes (both sexes). All costs associated with sexual attractiveness, mate search and encounter, courtship, post-copulatory mechanisms (if there are present) or oviposition associated costs are not present. I believe that if the sexes were compared but individuals who have already gone through a reproductive event could find differences in the trade-off that each sex displays between these reproductive activities and their immune function (immunocompetence).
488: Add citations in spiders
e.g., Ahtiainen, J. J., Alatalo, R.V., Kortet, R., & Rantala, M. J. (2005). A trade-off between
sexual signaling and immune function in a natural population of the drumming wolf spider
Hygrolycosa rubrofasciata. Journal of Evolutionary Biology, 18 (4), 985-991.
Calbacho-Rosa, L., Moreno-Garcia, M.A., Lanz-Mendoza, H., Peretti, A.V., & Córdoba-Aguilar,
A. (2012). Reproductive activities impair immunocompetence in Physocyclus dugesi (Araneae:
Pholcidae). The Journal of Arachnology, 40 (1), 18-23.
416-423 and 471-488: I suggest that the paragraphs be unified. Also, this paragraph should be simplified and rewritten more clearly and without repetition. Present the main results at the beginning of the paragraph and compare it with other taxa with similar results.
496-507: I never heard about the "quasi-terminal investment" and I would like to have some citation. The theory of "terminal investment" is about the strategy of each individual for multiple matings and trade-offs with future matings (i.e., the shift in allocation towards current reproduction with increasing age). Iteroparous or semelparous species can make a terminal investment if they invest more resources and energy in their last compared to the previous ones, it is an intra-individual approach that has to do with the age of the individual and the senescence. You could talk about "terminal investment" if you saw for example that the trade-off (somatic vs. reproductive) is different in the last mating regarding the first or towards the end of the reproductive season with respect to the beginning of the same. I think that here the concept is incorrectly handled.
508-513: This paragraph would place after 454.
541: Order the citations chronologically.

Figures and tables: in the captions add the level of significance considered.
Tables: unify number of decimals (table 1: 2-3 decimals, table 2: 3 decimals, table 3: 2-3 decimals). I recommend 3 decimals.
Table 1: in the title of the caption add "effect of developmental stage AND SEX". In "for discrete predictors" change to "for categorical predictors".
Table 3: unify "" or '' throughout the manuscript. Better "p-value" instead of P (to match with t-value). To unify with previous tables (1) put in the last row: Season (S - LS) for example and add a reference in table caption (S: spring females, LS: late summer females).

References: scientific names must be in italics.

---

## Round 0.3 · accepted · Accept

Dear Dr. Rádai and colleagues:

Thanks for revising your manuscript, and for addressing the concerns raised by the reviewers. I now believe that your manuscript is suitable for publication. Congratulations! I look forward to seeing this work in print, and I anticipate it being an important resource for research on spider antimicrobial immune responses.

Thanks again for choosing PeerJ to publish such important work.

-joe